# CeO_2_-rGO Composites for Photocatalytic H_2_ Evolution by Glycerol Photoreforming

**DOI:** 10.3390/ma16020747

**Published:** 2023-01-12

**Authors:** Stefano Andrea Balsamo, Eleonora La Greca, Marta Calà Pizzapilo, Salvatore Sciré, Roberto Fiorenza

**Affiliations:** 1Department of Chemical Sciences, University of Catania, Viale A. Doria 6, 95125 Catania, Italy; 2Institute for The Study of Nanostructured Materials (ISMN)-CNR, Via Ugo La Malfa 153, 90146 Palermo, Italy

**Keywords:** hydrogen, photoreforming, CeO_2_, graphene oxide, gold nanoparticles

## Abstract

The interaction between CeO_2_-GO or CeO_2_-rGO and gold as co-catalysts were here investigated for solar H_2_ production by photoreforming of glycerol. The materials were prepared by a solar photoreduction/deposition method, where in addition to the activation of CeO_2_ the excited electrons were able to reduce the gold precursor to metallic gold and the GO into rGO. The presence of gold was fundamental to boost the H_2_ production, whereas the GO or the rGO extended the visible-light activity of cerium oxide (as confirmed by UV-DRS). Furthermore, the strong interaction between CeO_2_ and Au (verified by XPS and TEM) led to good stability of the CeO_2_-rGO-Au sample with the evolved H_2_ that increased during five consecutive runs of glycerol photoreforming. This catalytic behaviour was ascribed to the progressive reduction of GO into rGO, as shown by Raman measurements of the photocatalytic runs. The good charge carrier separation obtained with the CeO_2_-rGO-Au system allowed the simultaneous production of H_2_ and reduction of GO in the course of the photoreforming reaction. These peculiar features exhibited by these unconventional photocatalysts are promising to propose new solar-light-driven photocatalysts for green hydrogen production.

## 1. Introduction

The development of the hydrogen economy is currently a hot topic highlighted by the recent politic and energy scenario and promoted by increasing interest in the circular economy. In this view, H_2_ is the ideal energy vector because it can be produced with several sustainable technologies and obtaining, as the main by-products of its combustion, water and CO_2_ that can be easily integrated in circular approaches (as the “blue hydrogen” where the obtained CO_2_ is captured and stored [1]). In the vision of an even more decarbonized economy, the production of hydrogen in fully sustainable ways (i.e., “green hydrogen”) is mandatory, together with the use of uncritical materials [2,3]. The solar photoreforming of sustainable organic substrates can be considered a well-performing and green process that allows the production of hydrogen from biomass or waste, valorising them as raw materials and using the solar energy as a renewable energy source [2]. To increase the hydrogen evolution from photoreforming, the suitable selection of photocatalysts that are able to absorb the solar radiation and efficiently oxidize the organic substrates acting as hole scavengers is necessary. 

Therefore, the employed photocatalysts (usually semiconductors) should own the conduction band more negative of the H^+^/H_2_ potential and the valence band more positive of the H_2_O/O_2_ couple and of the oxidation potentials of organic compounds [4]. For the photoreforming reaction, TiO_2_-based photocatalysts are the most investigated materials [5]. However, just two years ago the EU added titanium to the critical raw materials list [6]. In this contest for the overall sustainability of the process, the investigation of new photocatalysts is necessary.

Among these, in recent years cerium oxide (CeO_2_, ceria) has emerged as an interesting alternative to TiO_2_ or ZnO due to its redox properties, suitable band position for the photocatalytic H_2_ evolution and the lower bandgap (2.7–2.9 eV) with respect to ZnO and TiO_2_ [7,8,9,10]. To enhance the H_2_ production with the CeO_2_-based materials, chemical and/or structural modifications are necessary. The addition of carbonaceous materials, such as graphene oxide (GO) or reduced graphene oxide (rGO), can increase the separation between the photoproduced e^−^/h^+^ charge carriers and the absorption of the visible part of the solar radiation [11,12,13,14].

To the best of our knowledge, the study of the interaction between the CeO_2_ and the GO and/or rGO structures is poorly investigated in the literature for the solar photoreforming reaction. For this reason, we examined CeO_2_-GO and CeO_2_-rGO composites prepared with easy and green procedures also employing an in situ photoreduction step that allowed the progressive GO reduction into rGO with the contextual evolution of hydrogen. These interesting strategies allowed us to obtain versatile and unconventional photocatalysts that can be proposed as new and green materials for the sustainable H_2_ production. The main objectives of the work are (i) the investigation of the performance in the photocatalytic H_2_ production of unconventional CeO_2_-Au-GO and CeO_2_-Au-rGo composites; (ii) the study of the modification of the chemico-physical features of CeO_2_ due to the addition of gold and/or GO and rGO and (iii) the application of an in situ one-step solar photoreduction method for CeO_2_-based materials.

As the renewable organic substrate for the photoreforming reaction, we chose glycerol due to the growing interest in its valorisation, being obtained in large amounts as secondary product of biodiesel [15,16]. In this contest, the development of sustainable processes, such as H_2_ production by glycerol photoreforming, can have a double positive effect: (a) the increase in the viability of the biodiesel market with the contextual production of hydrogen and (b) the development of green technologies to exploit the biomass, the glycerol being a biomass derivate. Furthermore, compared to the other organic substrates produced from biomass and used as sacrificial agents for the photoreforming reaction, glycerol leads to easier hydrogen formation, with seven moles of H_2_ evolved for each glycerol molecule (reaction (1)) [17,18]:C_3_H_8_O_3_ + 3H_2_O → 7H_2_ + 3CO_2_(1)

## 2. Materials and Methods

### 2.1. Sample Preparation

Bare CeO_2_ was prepared starting from Ce(NO_3_)_3_·6H_2_O (Fluka, Buchs, Switzerland) solubilized in a 0.02 M CTAB solution. The pH was kept above 8, slowly dropping to KOH 1 M. The obtained suspension was stirred at 80 °C for 3 h and then left to digest at room temperature for 1 day. Then, the slurry was centrifuged at 8000 rpm for 15 min several times, washed with deionized water and then dried overnight in oven at 80 °C. Finally, the powders were calcined in air at 350 °C for 3 h. The same method was followed for the synthesis of another CeO_2_ sample synthetized without any templating agent, coded as CeO_2_-noCTAB.

CeO_2_-rGO and CeO_2_-rGO-Au samples were obtained following a slight modified method proposed in our previous works [12]. Briefly, 250 mg of the as-prepared ceria with CTAB was added to 37.5 mL of absolute ethanol, together with the stoichiometric volume of HAuCl_4_ in aqueous solution in order to obtain the 1 wt% amount of gold and adding previously sonicated (1 h at room temperature) aliquots of GO in aqueous solution. This slurry was prepared in a batch reactor and purged with nitrogen for 30 min in order to remove all of the oxygen. Then, the slurry was irradiated for 2 h with a solar simulator (Newport, equipped with a 150 W xenon lamp, photon flux ca. equal to 1.9 × 10^−7^ Einstein s^−1^ measured with ferrioxalate actinometer). Finally, the powders were dried overnight under vacuum at 60 °C. This temperature was chosen in order to not alter the reduction degree of the GO. The sample CeO_2_-GO was prepared in the same way described before but without any irradiation and without the addition of the gold precursor. The sample CeO_2_-Au was prepared without adding the GO in the reaction slurry, whereas the CeO_2_-GO-Au catalysts were obtained starting from the as-prepared CeO_2_-Au and adding without irradiation the GO. In addition, these powders were dried overnight under vacuum at 60 °C.

### 2.2. Photocatalytic Tests

The photocatalytic hydrogen production was conducted under solar irradiation, with a suspension of 1 mg mL^−1^ of catalyst in 10% (*v*/*v*) solution of glycerol in MilliQ water. In detail, a jacketed two-side neck batch reactor with an inner free volume of 68 mL was filled with 5 mg of catalyst and 5 mL of the glycerol solution. The reaction mixture was then purged for 30 min with N_2_. The solar irradiation was performed with a Newport solar simulator equipped with a 150 W xenon lamp (photon flux ca. equal to 1.9 × 10^−7^ Einstein s^−1^) thanks to an optical fibre enclosed in a cylindrical tube inside the reactor through the central neck. After 3 h of irradiation, 1 mL of the reaction gases was collected with a gastight syringe and analysed by gas chromatography with a thermal conductivity detector (GC-TCD) using an Agilent 7890A equipped with a Supelco Carboxen 1010 column.

### 2.3. Characterization Techniques

UV-vis DRS measurements were carried out with a Perkin Elmer Lambda 650 S UV/VIS spectrophotometer equipped with a 60 mm integrating sphere and using BaSO_4_ as reference material. The optical band gap was estimated by the TAUC PLOT method applying the Kubelka–Munk function.

FTIR was carried out with a Perkin-Elmer Spectrum Two FT-IR Spectrometer.

Raman spectra were acquired by a JASCO NRS-5500 Raman instrument equipped with a green laser (532 nm, power 1.7 mW) at 10s exposure and acquiring with a CCD detector (temperature 203 K) an L1800 grating with slit setting at 100 × 1000 µm (resolution 4 cm^−1^).

XPS data were recorded on pellets (4 mm diameter, 0.5 mm thick) after outgassing the samples to a pressure below 2 × 10^−8^ Torr at 150 °C. A Leibold–Heraeus LHS10 spectrometer (SPECS, Berlin, Germany) was adopted with an Al Kα X-ray source (hν = 1486.6 eV) at 120 W and 30 mA, using C (1s) or Au (4f_7/2_) as the binding energy reference (284.8 eV and 84.0 eV, respectively) and analysing the spectra with CasaXPS software.

Textural properties were characterized by N_2_ adsorption–desorption measurements carried out by a Micromeritics Tristar II Plus 3020 (Micromeritics Instrument Corp. Norcross, GA, USA) applying the Brunauer–Emmet–Teller (BET) and the Barret, Joyner and Halenda (BJH) methods in order to estimate the surface area and the pore size distribution, respectively.

Finally, microstructural analysis was accomplished by TEM using a Thermo Scientific Talos F200i S/TEM microscope operating at 200 kV. Both conventional TEM and STEM mode using high angle annular dark field (HAADF) imaging were performed. Moreover, elemental mapping was carried out using energy-dispersive X-ray spectroscopy (EDS) to study the elemental distribution throughout different areas of the sample. EDS maps were acquired using the VELOX software with a dwell time of 100 µs and 256 × 256 frame size.

## 3. Results

### 3.1. Photocatalytic Hydrogen Evolution

The CeO_2_ prepared with the CTAB was used in all the photocatalytic tests. Indeed, it was not possible to determine the H_2_ produced with the CeO_2_-noCTAB sample due to the very low amount (lower than the sensitivity of GC), whereas the sample prepared with the surfactant showed a production of 5 μmol g_cat_^−1^ h^−1^. Then, preliminary tests were carried out in order to evaluate the best rGO content added to the cerium dioxide as a photocatalyst.

Figure 1 reports the H_2_ evolution results over CeO_2_-based samples with 0.5 wt%, 1 wt% and 2 wt% of rGO.

The samples with 0.5 wt% and 1 wt% of rGO show similar activity, with production rates of around 225 μmol g_cat_^−1^ h^−1^ and 210 μmol g_cat_^−1^ h^−1^, respectively. Slightly lower performances were obtained with the 2 wt% sample. Considering the small difference within the error bars between the 0.5 wt% and the 1 wt% samples, the latter was chosen as the reference amount for this work in order to facilitate the catalysts characterization.

In Figure 2, the photocatalytic activity data for the other examined samples are reported. Bare CeO_2_ shows very low activity, with a production rate of 5 μmol g_cat_^−1^ h^−1^, slightly enhanced by the addition of GO or rGO (up to 20 μmol g_cat_^−1^ h^−1^ for both the modified samples). As reported by Christoforidis and Fornasiero [4], the activity of a semiconductor without any co-catalyst is very low, especially for materials like CeO_2_ that have a charge recombination rate faster than TiO_2_. In this context, the addition of GO on CeO_2_ allowed us to obtain only a slight increase in the photocatalytic activity. However, as expected, the addition of a metallic co-catalyst increases the activity up to 90 μmol g_cat_^−1^ h^−1^ for the CeO_2_-Au sample and to 210 μmol g_cat_^−1^ h^−1^ in the presence of GO or rGO.

It is evident how the electron-sink effect of the carbonaceous material is helped by the beneficial effects of gold as a co-catalyst. This behaviour suggests an important role of the gold in the electron transport mechanism. This is clear because the hydrogen production is not affected by the status of the graphene oxide (GO or rGO), with the activities of CeO_2_-GO and CeO_2_-rGO being similar, as well as those of CeO_2_-GO-Au and CeO_2_-rGO-Au. This can be ascribed to the weak photoreduction degree of GO caused by the poor photocatalytic features of CeO_2_ under solar irradiation. 

However, the photocatalytic properties of ceria can be better exploited after progressive runs. Figure 3 shows the results of five subsequent runs of the CeO_2_-rGO-Au catalyst. Interestingly, an increment of 60 μmol g_cat_^−1^ h^−1^ was observed after five runs, attributed to the progressive rGO reduction promoted by the increase in the time of solar irradiation (3 h for every run). 

### 3.2. Photocatalyst Characterizations

Figure 4 shows the UV-vis DRS spectra of the examined samples and their relative TAUC plots for the estimation of the optical band gaps. The absorption spectrum of the bare ceria prepared without CTAB as the templating agent is reported in panel (a). It is possible to note, according to the literature, that the presence of CTAB induces a slight bathochromic shift in the absorption band (i.e., decrease in the optical band gap) [19], and this is another positive feature of the use of the templating agent in the CeO_2_ synthesis. In addition, the presence of graphene oxide does not give evident variation in the band onset. In the gold-containing samples, it is possible to distinguish the signal at 556 nm due to the plasmon of the metal. According to the surface plasmon resonance (SPR) effect theory, due to the lower dielectric constant of CeO_2_ compared to TiO_2_, the position of this signal is shifted to higher wavelengths with respect to the TiO_2_-based samples [20,21,22]. The figure in panel (b) illustrates the TAUC plots for the previously discussed samples. The smallest band gaps are obtained with the CeO_2_-rGO-Au and CeO_2_-GO-Au samples (2.56 eV). Analogously, no substantial differences were detected between the CeO_2_-GO and the CeO_2_-rGO samples (2.72 eV and 2.71 eV, respectively).

It must be noted that in order to verify the removal of the CTAB templating agent after the synthesis, a calcination temperature of 350 °C was chosen for all the samples [7]. Appendix A reports the FTIR spectra for the bare cerium dioxide before and after the calcination. It is evident how the signals related to the CTAB (the asymmetric and symmetric C-CH_2_ stretching vibrations of the methylene chains at 2919 cm^−1^ and 2851 cm^−1^, and the weak signal at 3011 cm^−1^ assigned to the N-CH_3_ stretching of the methyl group and their wagging vibrations in the intense and sharp peak at 1390 cm^−1^) disappear after the calcining treatment, confirming the elimination of the surfactant [19].

Figure 5 shows the Raman spectra of the investigated samples. The band at around 460 cm^−1^ is the F_2g_ signal related to the fluorite-phase skeletal vibrations. No significant variations in its position were detected, thus the ceria structure was not modified due to the addition of GO and/or gold. In all the spectra the weak signal at 600 cm^−1^ due to Frenkel defects on the bulk is evident. For this reason, a modest quantity of Ce^3+^ sites are expected [23].

To investigate the occurrence of the reduction process from GO to rGO, we compared the bands of the rGO-containing samples with the bare GO spectrum (showed in Appendix A). All the samples show the typical bands of this material. The most important are surely those located at 1344 cm^−1^ and 1600 cm^−1^ (D and G band, respectively). These two signals are important because thanks to them it is possible to estimate the reduction degree of the carbonaceous material: the higher their ratio is (I_D_/I_G_), the higher the amount of disordered carbons [12,24]. The parameters related to these bands are reported in Table 1.

The I_D_/I_G_ ratio increases from 0.88 to 0.99 with the addition of the ceria. As reported in the literature, this is due to the interaction between the terminal oxygens of ceria and the graphene oxide layers [25]. Nevertheless, this value does not change after a photoreduction treatment in the presence or absence of gold. After five runs, it reaches a value of 1.02. However, a slight reduction trend can be found looking at the FWHM values that slightly go down in the reduced samples. Indeed, the reduction degree can also be evaluated as a separation between the two bands, often approximated with the FWHM values: the lower they are, the higher the reduction degree [26]. A slight decrease can be noted after the reduction treatment (136-76 vs. 188-90 in the CeO_2_-rGO and CeO_2_-GO, respectively), in addition to an increase in the presence of gold (159-84 for CeO_2_-rGO-Au) and once again a decrease after a long irradiation (136-70 for the CeO_2_-rGO-Au after 5 runs). Considering these behaviours, it can be concluded that the photocatalytic features of cerium dioxide under solar irradiation are not enough to obtain an efficient GO reduction, which starts to be progressively reduced after long irradiation steps and in situ during the photocatalytic hydrogen evolution. Finally, at higher Raman shift the second-order bands due to the carbon are present at 2660 cm^−1^ (2D band), at 2930 cm^−1^ (D + D’ band) and at 3192 cm^−1^ (2D’ band). The 2D band is normally forbidden and becomes visible with defective structures. The D + D’ band is ascribed to the combination of phonons with different momenta, so the presence of defects is necessary to allow the transition according to the selection rules [27]. Instead, the 2D’ band is the only one permitted by the selection rules [26].

The textural properties of the samples were investigated by N_2_ adsorption–desorption isotherms, evaluating the BET surface area and the BJH pore size distribution (Appendix A). All the isotherms are of type IV, confirming the mesoporous nature of the CeO_2_-based samples. Each curve shows an H2-type hysteresis typical of ink bottle pores, as widely reported for metal oxides [28]. The BET surface area and the mean pores size are summarized in Table 2.

There is a general decrease in surface area compared to the bare CeO_2_ (about 10 m^2^ g^−1^) after the photoreduction of gold and GO. This can be reasonably ascribed to the occurrence of a partial agglomeration of the CeO_2_ particles due to the further thermal treatment (drying at 60 °C) and the 2 h of solar irradiation [29,30]. Moreover, the intercalation of the CeO_2_-Au nanoparticles with the graphene oxide sheets induced pores modification [12]. Indeed, the mean pore size was 8 nm in the CeO_2_-Au sample, whereas it was 17 and 18 nm for the CeO_2_-GO-Au and CeO_2_-rGO-Au samples, respectively. This can be due to the presence of interstitial spaces between the nanoparticles and the graphene oxide sheets [31].

To analyse the morphology of the prepared samples, we performed TEM microstructural measures of the CeO_2_-rGO-Au as the representative sample (Figure 6).

In Figure 6a it is possible to note the morphology of the nanoparticles modified with rGO. In this image, the elemental analyses were carried out to examine the elements’ configurations. The most remarkable image (Figure 6c) shows that the gold is homogeneously dispersed on the catalyst, whereas in Figure 6e it is possible to note that the CeO_2_-Au nanoparticles are preferentially deposited on the rGO sheets, distinguishable from the delineated edges.

Finally, the Ce 3d and O 1s XPS analyses for the CeO_2_-rGO-Au sample are reported in Figure 7, while the XPS analyses of the other samples are reported in Appendix A.

Panel (a) reports the Ce analysis of the CeO_2_-rGO-Au sample. In this region, all the characteristic peaks of the cerium are present, including the Ce^3+^-related signals at 885 and 903 eV [29,32]. The Ce^3+^/Ce^4+^ ratio is a parameter that gives us an idea of the reduction processes and/or defectivity of the material. This value passes from 0.42 of bare CeO_2_ (spectrum showed in the Appendix A) to 0.33 of the CeO_2_-rGO-Au sample shown in panel (a), indicating a slight overall oxidation of the material. This behaviour can be fairly understood if we assume that the photoreduction process rearranges the chemical structure, eliminating the defective Ce^3+^ sites. Indeed, it is reasonably assumed that a reduction of the GO corresponds to an overall oxidation of the Ce^3+^ species into Ce^4+^. Panel (b) shows the O 1s spectrum of the same sample. Here the signals are assigned to five different components: Ce^IV^-O and Ce^IV^-OH (529.5 eV and 531 eV), adsorbed carbonates on the surface (532 eV) and Ce^III^-O and Ce^III^-OH (533 eV and 534 eV) [33]. The same signals are found on the bare CeO_2_ sample (Appendix A). Regarding the other features, in the C 1s region (Appendix A), the three principal signals are related to the C-C, the C-O and the C=O of the reduced graphene oxide [32]. In the Au 4f region (Appendix A), there are the two signals attributed to the Au 4f_7/2_ and 4f_5/2_ at 84.0 eV and 87.5 eV, respectively, typical of metallic gold. This pointed to that during the photoreduction process an efficient photoreduction of the gold precursors occurred, explaining the increase in the photocatalytic activity verified for all the gold-based samples.

## 4. Discussion

From the data reported in the previous sections it is clear that all the components of the proposed photocatalytic composites (CeO_2_, GO or rGO and Au) contributed to both the photoreduction/deposition preparation method adopted and the H_2_ evolution by glycerol photoreforming. Indeed, the ceria was the photoactive semiconductor that, due to the solar irradiation, allowed the formation of the e^−^/h^+^ pairs, able to start the involved reactions. The addition of GO permitted, instead, to extend the photo-response to a larger part of the visible portion of the solar incident light, as confirmed by the optical band gap decrease from 2.75 eV of bare ceria (λ_excitation_ ≤ 451 nm) to 2.56 eV of the CeO_2_-GO-Au sample (λ_excitation_ ≤ 484 nm). A further exploitation of the visible light was endorsed by the presence of gold due to its surface plasmon resonance effect (activated at 556 nm, as detected by the UV-DRS), which induced the gold electrons to participate in the photoreforming reactions, thus increasing the photocatalytic activity. Notwithstanding that the photocatalytic features of CeO_2_ were poor compared to TiO_2_ or ZnO [34,35], the strong interaction between gold and ceria [36,37], also favoured by the presence of the GO sheets, granted a good photodeposition/reduction of Au (confirmed by TEM and XPS).

It is important to note that the proposed photoreduction preparation required the occurrence of similar reactions of the photoreforming process. The first reaction (2) is the formation of the charge carriers, and subsequently the holes (h^+^) in the valence band (VB) of ceria react with the sacrificial agent (ethanol in the case of the photoreduction method, glycerol in the photoreforming reaction) to promote the charge carrier separation (3). In this way the excited electrons in the conduction band (CB) of ceria are able to reduce the gold precursor (4) and/or the GO to rGO (5) (Figure 8).
CeO_2_ + hν(solar) → CeO_2_(e^−^ + h^+^)(2)
CeO_2_(e^−^ + h^+^) + s.agent → CeO_2_(e^−^) + s.agent^+^(3)
CeO_2_(e^−^) + Au^3+^ → CeO_2_ + Au(4)
CeO_2_(e^−^) + GO → CeO_2_+ rGO(5)

In the case of photoreforming after reactions (2) and (3), the evolution of H_2_ was obtained by means of the same excited electrons of the CB of ceria and by the presence of glycerol (6) and (7).
C_3_H_8_O_3_ + 3H_2_O + 2h^+^_CeO_2__ → 3CO_2_ + 6H_2_+2H^+^(6)
CeO_2_(e^−^) + 2H^+^ → CeO_2_+ H_2_(7)

Interestingly, while for the photoreforming the presence of gold resulted in an important increase in H_2_ evolution, the addition of GO or rGO had the same effect (Figure 2). As detected by the Raman measurements, reaction (5) was favoured after subsequent runs of continuous solar photoreforming reaction, therefore the reduction of GO into rGO further proceeded together to the reactions (6) and (7). As a result, the activity of the CeO_2_-rGO-Au increased (Figure 3) from 210 μmol g_cat_^−1^ h^−1^ in the first run to 270 μmol g_cat_^−1^ h^−1^ in the fifth run. This behaviour was different compared to that reported for conventional photocatalysts [38,39], which after different consecutive runs maintained the activity or were deactivated due to the occurrence of the e^−^/h^+^ recombination.

This is a fascinating result, with the obtained H_2_ that was higher than other CeO_2_-based systems reported in the literature for solar photoreforming (Table 3), even though the different experimental set-ups employed by the various research groups must be considered.

This one-pot material synthesis with a progressive performance improvement during the photocatalytic reaction can open new ways of designing solar-light-driven photocatalysts that can enhance their photoactivity in situ through the reaction. This versatile approach can also be useful for possible scale-up applications.

## 5. Conclusions

A solar one-pot photoreduction method was investigated in this work for the synthesis of cerium dioxide-graphene oxide-based systems. The obtained materials were used to catalyse the solar glycerol photoreforming for green H_2_ production. The photocatalytic activity data highlighted an increase in the H_2_ evolution due to the addition of graphene oxide or reduced graphene oxide to ceria, further boosted by the presence of gold as a co-catalyst. Furthermore, the strong interaction between this metal and CeO_2_ led to excellent stability performances of the CeO_2_-rGO-Au sample, with a progressive increase in the H_2_ production over the runs ascribed to a contextual enhancement of the amount of GO reduced into rGO. The presence of gold is fundamental for the enhancement of the H_2_ production due to the exploitation of the surface plasmon resonance effect with a contextual better use of the visible light portion of the solar irradiation, whereas the presence of GO and rGO sheets promoted a good dispersion of the Au nanoparticles. In conclusion, in the glycerol photoreforming, the synergism between the photocatalytic features of CeO_2_, the visible-light-induced properties of gold and the presence of GO and rGO allowed to obtain versatile photocatalysts activated by solar irradiation with an improved photoactivity induced by the continuous light absorption and by the subsequent photocatalytic runs. This can be a starting point to propose new solar-driven photocatalysts for sustainable hydrogen production.

## Figures and Tables

**Figure 1 materials-16-00747-f001:**
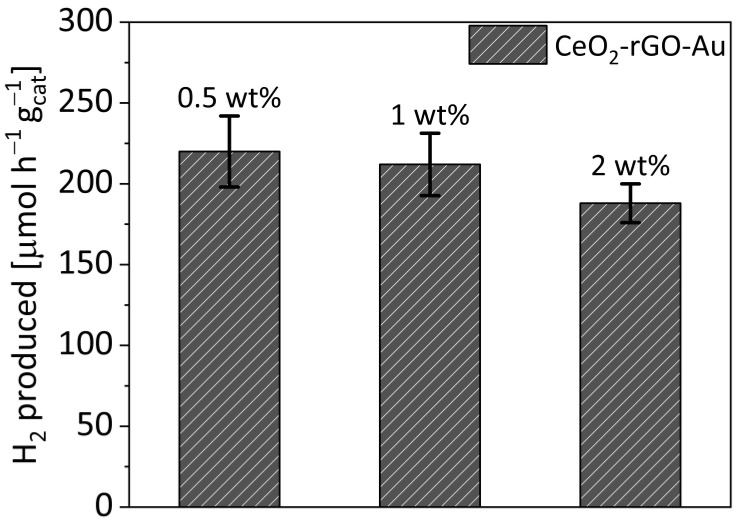
H_2_ evolution considering the different rGO content on CeO_2_-based samples. All the samples contain gold at 1 wt% as co-catalyst.

**Figure 2 materials-16-00747-f002:**
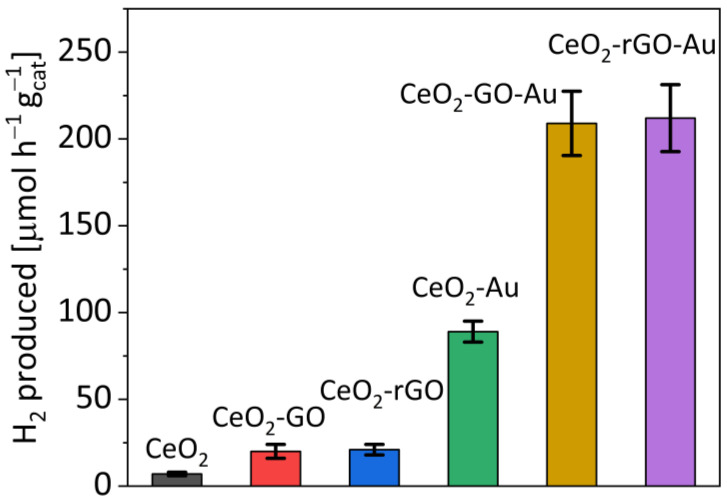
H2 production over the different CeO_2_-based samples.

**Figure 3 materials-16-00747-f003:**
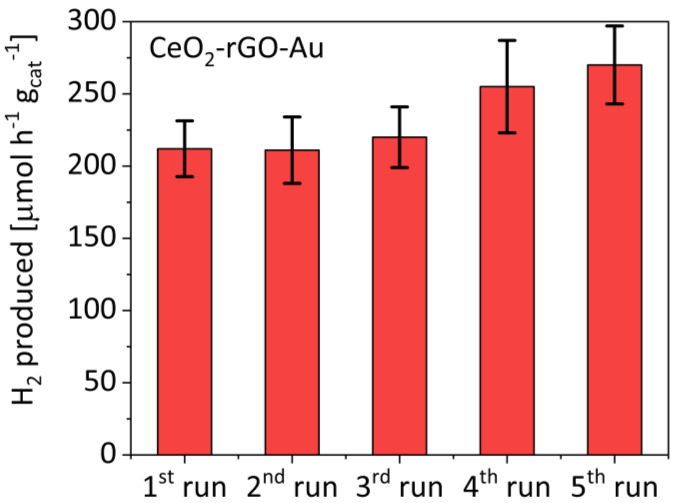
Photocatalytic activity of the CeO_2_-rGO-Au sample over five subsequent runs.

**Figure 4 materials-16-00747-f004:**
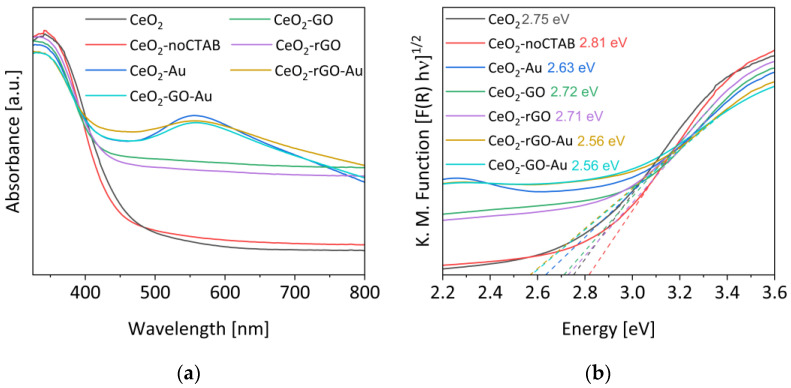
(**a**) UV-DRS of the samples and (**b**) their TAUC plots with the corresponding linear fitting (dashed lines) for the estimation of the band gap.

**Figure 5 materials-16-00747-f005:**
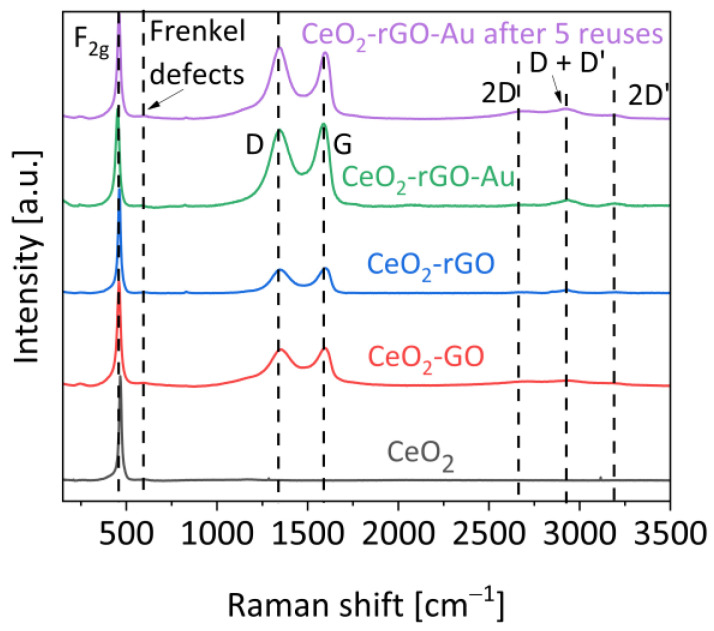
Raman spectra of the samples. The discussed signals are highlighted by dashed lines.

**Figure 6 materials-16-00747-f006:**
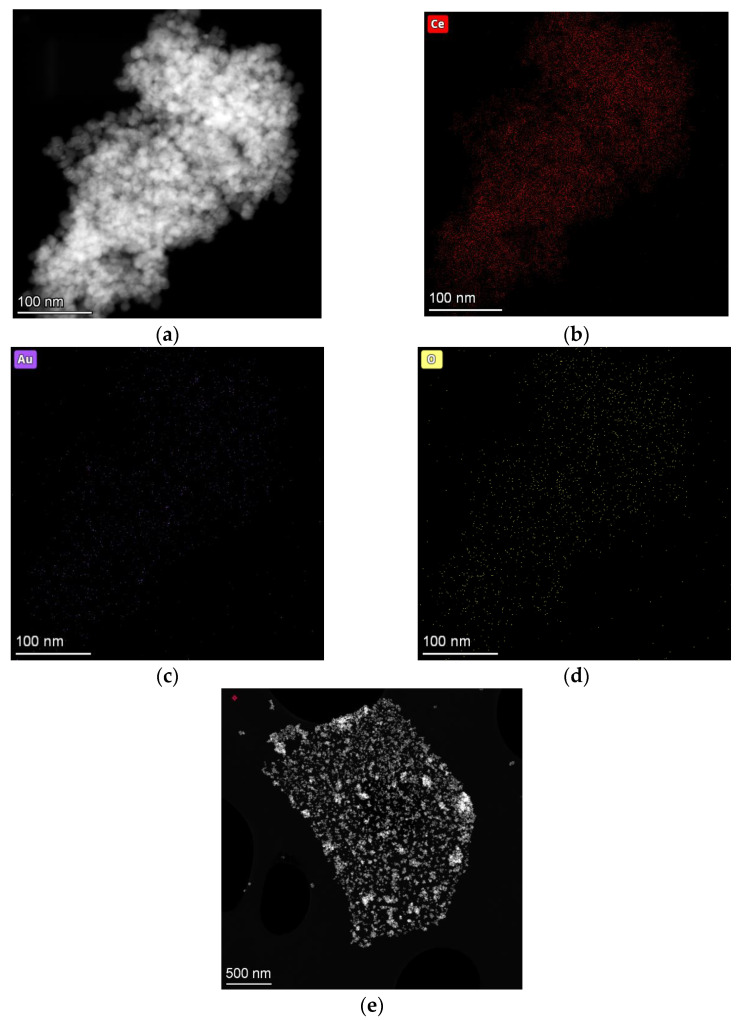
STEM micrograph of (**a**) CeO_2_-rGO-Au sample and its elemental mapping; (**b**) Ce; (**c**) Au and (**d**) O. Panel (**e**) is a micrograph in which it is evident how the CeO_2_-Au nanoparticles are deposited in a GO sheet distinguishable for the delineated edges.

**Figure 7 materials-16-00747-f007:**
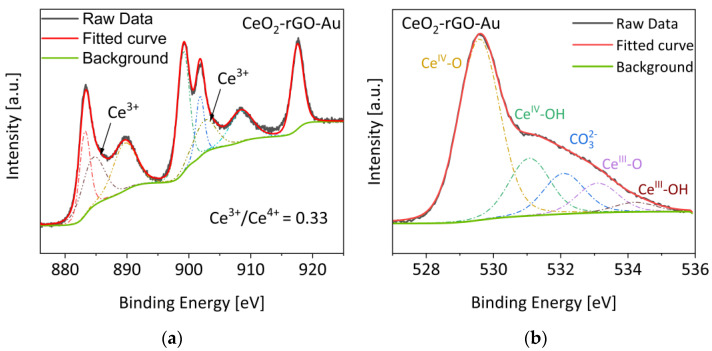
(**a**) Ce 3d and (**b**) O 1s XPS spectra of CeO_2_-rGO-Au. The deconvolution curves of the single components are depicted by dashed/dotted lines.

**Figure 8 materials-16-00747-f008:**
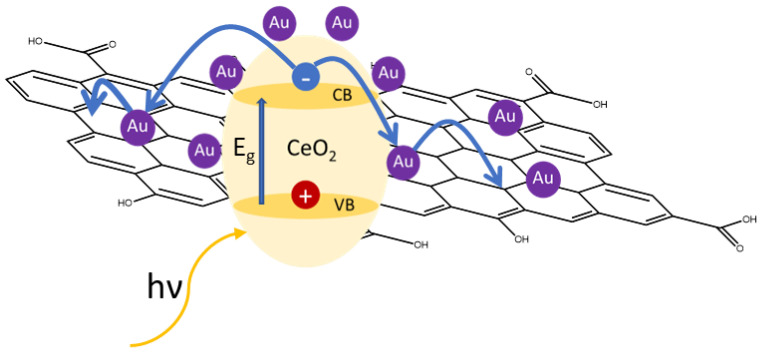
Representative scheme of the interactions in the sample CeO_2_-rGO-Au.

**Table 1 materials-16-00747-t001:** Raman parameters related to the D and G bands.

Sample	I_D_/I_G_	FWHM (cm^−1^)
GO	0.88	166_D_, 83_G_
CeO_2_-GO	0.99	188_D_, 90_G_
CeO_2_-rGO	0.99	136_D_, 76_G_
CeO_2_-rGO-Au	0.99	159_D_, 84_G_
CeO_2_-rGO-Au after 5 runs	1.02	130_D_, 70_G_

**Table 2 materials-16-00747-t002:** Textural properties: type of N_2_ isotherms, BET surface area and mean pore size.

Sample	Isotherm	S_BET_ (m^2^ g^−1^)	d_P_ (nm)
CeO_2_	IV; H2	81 ± 2	24
CeO_2_-Au	IV; H2	72 ± 2	8
CeO_2_-GO-Au	IV; H2	69 ± 2	17
CeO_2_-rGO-Au	IV; H2	70 ± 2	18

**Table 3 materials-16-00747-t003:** Comparison among several CeO_2_-based systems used for the solar photocatalytic H_2_ production.

Sample	Synthesis	Photoreforming Conditions	Irradiation Source	H_2_ Evolution (μmol/g_cat_ h)	Ref.
CeO_2_-rGO-Au (after 5 runs)	Photoreduction	10% (*v*/*v*) solution of glycerol in MilliQ water _G_	150 W Xe lamp solar simulator	270	this work
CdS/Pt/WO_3_-CeO_x_	Precipitation	In situ material synthesis and H_2_ evolution	300 W Xe lamp, with a 420 nm cutoff filter	50	[40]
CeO_2_/C quantum dots	Bio-template strategy	100 mL of water containing 10 vol% methanol	300 W, 50 mW/cm^2^ Xe lamp	60	[25]
CeO_2_/MoS_2_	Hydrothermal	100 mL aqueous solution containing 0.3 M Na_2_SO_3_/Na_2_S	150 W xenon arc lamp (>400 nm)	125	[41]
C_3_N_4_/CeCO_3_OH/CeO_2_	Hydrothermal	80 mL water including 10 vol% TEOA	AM 1.5G solar light	40	[42]
CeO_2_ nanorods	Hydrothermal	100 mL of an aqueous solution containing 0.43 M Na_2_S mixed with 0.50 M Na_2_SO_3_	300 W Xe lamp	5	[43]

## Data Availability

The data are available on request to the corresponding author.

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
