# Peer review of "CeO_2_-rGO Composites for Photocatalytic H_2_ Evolution by Glycerol Photoreforming"

_materials, 2023, doi:10.3390/ma16020747_

Round 1
Reviewer 1 Report
The work of Stefano Andrea Balsamo et al. about CeO2-rGO composites for photocatalytic H2 evolution is quite interesting. Therefore, I recommend to submit this work to journal. However, some major issues have to addressed before submission:
1. The typo error should be checked before submission.
2. The role/importance of glycerol photoreforming should be introduced in the Introduction section detaily.
3. A brief illustration/figure of morphology or the interaction between components in composite could be improved the quality of this work.
4. In the Raman sepctra study, authors wrote that "...the higher is their ratio (ID/IG) the higher is the reduction degree". Indeed, "higher ID/IG ratio" means higher amount of disordered carbon (i.e., sp3 carbon), that is another representation of oxidated carbon. In this viewpoint, the increase should be assigned to oxidation level, not "reduction degree". If this assumption is not correct, there is another assumption that sp2 carbon would react with other elements to generate sp3 carbon, or the amount of edge carbon in graphene would be increased. However, the authors also wrote that "the ceria structure was not modified due to the addition of GO and/or gold". ᅟHence, a reasonable explanation must be proposed.
5. The explanation of BET is not reasonable. In general, the specific surface area and pore size distribution represent a contrast. But in this work, it is not. Does the intercalaction of Au nanoparticles on/in CeO2 reduces its BET surface area? This assumption bases on an imagination that Au nanoparticles perfectly block the pores of CeO2 nanoparticles. This is technically incorrect. And if Au nanoparticles is partly block the pores, the BET surface area have to be increased.
Reviewer 2 Report
Comments to the Authors
In this manuscript authors examined CeO2-GO and CeO2-rGO composites prepared with easy and green procedures also employing an in-situ photoreduction step with the contextual evolution of hydrogen. This research has value for the researchers in the related areas. However, the paper needs improvement before acceptance for publication. My detailed comments are as follow:
1. In the introduction section authors should introduced following interesting article related to CeO2 and GO and their discussion:
a. doi.org/10.1007/s13233-019-7039-y and also other articles.
2. There are lots of typos and grammatical errors. Like “Error! Reference source not found”
3. STEM microgram should be included in the main text.
4. The writing of the objective of the work and conclusion section should be improved.
5. What is commercial viability of such prepared composites?
Round 2
Reviewer 1 Report
The quality of this work is now sufficient for publication.